# Working Conditions and Wellbeing among Prison Nurses during the COVID-19 Pandemic in Comparison to Community Nurses

**DOI:** 10.3390/ijerph191710955

**Published:** 2022-09-02

**Authors:** Megan Guardiano, Paul Boy, Grigoriy Shapirshteyn, Lisa Dobrozdravic, Liwei Chen, Haiou Yang, Wendie Robbins, Jian Li

**Affiliations:** 1School of Nursing, University of California Los Angeles, Los Angeles, CA 90095, USA; 2Quality Management Support Unit, Avenal State Prison, Avenal, CA 93204, USA; 3Department of Epidemiology, Fielding School of Public Health, University of California Los Angeles, Los Angeles, CA 90095, USA; 4Center for Occupational and Environmental Health, University of California Irvine, Irvine, CA 92617, USA; 5Department of Environmental Health Sciences, Fielding School of Public Health, University of California Los Angeles, Los Angeles, CA 90095, USA

**Keywords:** correctional nurses, occupational health, work conditions, mental health, prison, nurses, COVID-19

## Abstract

The psychological health and work challenges of nurses working in prisons during the COVID-19 pandemic are understudied. We evaluated the work and wellbeing characteristics of a California prison nurse group, with a comparison to those of a community nurse group. From May to November 2020, an online survey measured psychosocial and organizational work factors, sleep habits, psychological characteristics, COVID-19 impacts, and pre-pandemic recall among 62 prison nurses and 47 community nurses. Prison nurses had significantly longer work hours (54.73 ± 14.52, *p* < 0.0001), higher pandemic-related work demands, and less sleep hours (5.36 ± 1.30, *p* < 0.0001) than community nurses. Community nurses had significantly higher pandemic-related fear levels (work infection: *p* = 0.0115, general: *p* = 0.0025) and lower perceived personal protective equipment (PPE) supply (*p* = 0.0103). Between pre-pandemic and pandemic periods, both groups had significantly increased night shift assignments and decreased sleep hours, but the prison group had increased work hours. Although not statistically significant, both groups had high occupational stress and prevalence of post-traumatic stress symptoms. Our results indicate that prison nurses experienced work and wellbeing challenges during the pandemic. Future research and practice ought to address nurses’ workload, PPE, and psychological resources in correctional facilities and healthcare organizations.

## 1. Introduction

At the onset of the COVID-19 pandemic in 2020, an estimated 27,743 registered nurses (RNs) were actively working at United States (U.S.) correctional facilities, which include state prisons, as their primary employment, accounting for 0.8% of the U.S. RN population [1,2]. Throughout the pandemic, the National Institute for Occupational Safety and Health [3] and the National Academy of Medicine [4] have advocated for the occupational mental health and wellbeing of healthcare workers. The National Commission on Correctional Health Care, aligned with the Nurse Code of Ethics, supports correctional nurse health to optimize care for incarcerated individuals [5].

The job demands–resources (JD-R) theory provides a framework to demonstrate how the occupational environment and conditions may influence wellbeing among nurses working in prison settings. Initially, the occupational theory of JD-R related the concepts of elevated job demands and insufficient job resources with lower motivation and the outcome of burnout among workers [6]. The JD-R theory has been developed to broadly involve multiple [7] and various job demands, job resources, and worker health outcomes [8], applicable to any occupation [6]. Among nurses, evidence suggests that high psychological and physical job demands are related to the intention to quit nursing, emphasizing the relevance of the JD-R theory to this population [9]. Additional significant findings from research applications of the JD-R theory in the COVID-19 pandemic context indicate a negative association between the perception of work organization support and post-traumatic stress disorder symptoms in a sample of U.S. nurses working in COVID-19 hospital units [10]. As an extension of the JD-R theory to our study of prison nurses, the domain of job demands includes efforts, work hours, and the types of shift assignment and patient care. The job resources domain encompasses PPE supply and work-related rewards. The wellbeing outcome measures are the psychological characteristics of anxiety, depression, and post-traumatic stress symptoms. Thus, research applying this JD-R theory to correctional nurses working during the COVID-19 pandemic will further characterize the job demands, resources, and outcomes unique to this population and compounded by the pandemic.

Prior to the COVID-19 pandemic, published reviews related to nurses and other health professionals working in correctional facilities identified occupational stressors including security prioritization, conflicts, fear, job demands [11], burnout, stress [12], and secondary trauma [13]. Within North America, older U.S. studies have demonstrated moderate [14] and high [15] work-related mean stress levels among correctional nurses. Other North American pre-pandemic studies of correctional worker mental health in Canada have included nurses, but within healthcare worker subgroups [16,17].

A recent study of U.S. correctional workers during the COVID-19 pandemic found that correctional healthcare workers reporting any degree of depression, anxiety, burnout, and post-traumatic stress symptoms ranged from 37% to 50% [18]. However, this study was concentrated on correctional facilities located in eastern U.S. states [18]. To the best of our knowledge, there are no available scientific reports focused solely on prison nurses and their working conditions and wellbeing during the COVID-19 pandemic.

The prevalence of COVID-19 cases in U.S. correctional facilities has been significantly higher than that of the general population. Based on the facilities’ available reports, which vary in data quality, there were 42,107 COVID-19 cases among incarcerated individuals in U.S. federal and state prisons, a rate that was five-and-a-half-fold greater than that of the U.S. population, between March and June 2020 [19,20]. There were 13,781 documented or reported COVID-19 cases specifically in California correctional institutions, a rate that was on average over eight-and-a-half-fold greater than that of the aggregated Californian population between September and November 2020 [21].

Research on healthcare workers and nurses working through the COVID-19 pandemic has demonstrated elevated levels of occupational and psychological concerns. Studies conducted in Europe and Asia have identified elevated levels of occupational stress [22,23], insomnia, workload, anxiety, and depression [22]. Additionally, effort and over-commitment have been associated with anxiety and depression [24]. Literature reviews and meta-analyses of international healthcare workers have reinforced these individual study findings, with pooled prevalence rates ranging from 43% to 56.5% for stress, 40% to 44% for sleep issues [25,26,27], and 18.75% to 48% for post-traumatic stress [26,27,28].

Qualitative and quantitative research on U.S. nurses during the COVID-19 pandemic have recognized occupational challenges regarding patient care, increased workload, and inadequate personal protective equipment (PPE), as well as psychological outcomes including post-traumatic stress, depression, and anxiety [29,30,31,32]. However, these studies heavily focus on hospital settings, and are mostly concentrated in the U.S. Northeast, South, and Midwest [29,30]. A December 2020 national survey that provided state-specific data reported that the majority of California nurses felt exhausted, overwhelmed, and anxious, with 52% expressing neutrality or disagreement with the statement that their workplace valued employee safety and health [33]. Yet, there was minimal representation of correctional nurses in California [33]. In California correctional settings, the Legislative Analyst’s Office 2019 report acknowledged the California Department of Corrections and Rehabilitation’s (CDCR) use of mandatory overtime for nursing staff, despite previous state agreement to decrease this practice [34].

The underrepresentation of correctional nurses in California and the combined challenges intrinsic to the correctional work setting and to the COVID-19 pandemic warrant further investigation. To the best of our knowledge, this is the first study in the western U.S. region to exclusively target prison nurses and their working conditions and wellbeing in the context of the COVID-19 pandemic, and compared to a non-correctional worker group. This study aims to evaluate a group of California prison nurses and compare their work characteristics and wellbeing outcomes with those of a community nurse group.

## 2. Materials and Methods

### 2.1. Design

This cross-sectional study with convenience sampling utilized a one-time online survey to compare the occupational and wellbeing characteristics of nurse participants working in a prison (prison group) to those working in other clinical settings (community group).

### 2.2. Setting and Sample

Recruitment for the community group occurred through nursing organization websites during an approximate 1.5-month survey window between late May and early July 2020. Most of the community nurse participants were in California. The prison nurse group was subsequently enrolled through collaboration with healthcare administrators at a California state prison. The survey window for the prison nurse group was about two months, from early September to late November 2020. For both groups, eligible nurses had to have current paid employment in a healthcare setting since the start of the COVID-19 pandemic.

Informed consent was obtained from participants at the initiation of the online survey. Each participant received a USD 10 gift card incentive. This study was reviewed and approved by the University of California, Los Angeles Institutional Review Board (IRB#20-000804 and IRB#20-001440), and followed the Declaration of Helsinki guidelines, as well as the Strengthening the Reporting of Observational Studies in Epidemiology (STROBE) reporting guideline.

### 2.3. Measures

The “Survey of Nurses Work and Wellbeing during the COVID-19 Outbreak” was administered online to both groups of participants. The survey included validated instruments, Likert-type scales, visual analog scales, and numeric and free text responses to measure the working conditions and wellbeing of nurses before and during the pandemic (current context). Specifically, the survey focused on 3 domains of assessments on working conditions: psychosocial characteristics, organizational work characteristics, and the COVID-19 working characteristics. The survey also focused on 3 domains of assessments on psychological wellbeing, including sleep characteristics, psychological characteristics, and post-traumatic stress disorder. Pre-pandemic recall and current reports were requested for the variables of weekly work and sleep hours and night shift assignment. All other variables were one-time measurements.

#### 2.3.1. Psychosocial and Organizational Work Characteristics

The Effort–Reward Imbalance (ERI) scale was used to measure psychosocial factors at work, consisting of 10 items, 3 for effort, and 7 for reward [35,36,37]. The effort score ranges from 3 to 12 and the reward score ranges from 7 to 28, with high scores corresponding with high magnitudes of effort and reward [35,36,37]. The E–R ratio score ranges between 0.25 and 4.00, with scores above one suggesting high work stress [35,36,37]. The Cronbach’s alpha coefficients were 0.80 for the effort subscale, and 0.78 for the reward subscale. The ERI measure has been widely used among nurses in Europe [38], as well as healthcare workers, including nurses in the United States [39]. During the COVID-19 pandemic, several studies used the ERI for measuring work stress in frontline healthcare workers [22,23,24,40].

The organizational work characteristics included work years, as well as pre-pandemic and current (at the time of survey) measurements of weekly average work hours and frequency of night shift assignment.

#### 2.3.2. COVID-19 Characteristics

The COVID-19 working characteristics included perceptions of general pandemic-related fear, fear of infection, adequacy of PPE supply, magnitude of COVID-19 patient contact, requests or history of work department redeployment, and history of COVID-19 symptoms, testing, and diagnosis.

#### 2.3.3. Sleep Characteristics

Pre-pandemic and current weekly averages of sleep hours were collected. To measure insomnia levels, three items about sleep within the last month were obtained from the National Health and Nutritional Examination Survey [41]. Scores range between 0 and 12, with higher numbers associated with higher degrees of insomnia [41]. This insomnia measure had a Cronbach’s alpha of 0.83.

#### 2.3.4. Psychological Characteristics

The Patient Health Questionnaire-4 (PHQ-4) measured depression and anxiety. The PHQ-4 features two two-item subscales to measure depression and anxiety symptoms over the past month [42]. Each subscale’s score ranges from 0 to 6, with higher numbers relating to higher levels of depression and anxiety [42]. For both conditions, scores of 3 and above represent positive cases of depression and anxiety [42]. Both subscales were reliable, with a Cronbach’s alpha of 0.82 for depression and 0.9 for anxiety. Previous studies utilized this brief instrument during the COVID-19 pandemic among a hospital nurse sample in Romania [43], and a hospital nurse and nurse assistant sample in the United States [29].

Post-Traumatic Stress Disorder (PTSD) symptoms over the past month were measured with a six-item screening instrument [44]. Scores range from 6 to 30, with elevated scores reflecting elevated PTSD symptoms [44]. A score of 14 or above indicates PTSD [44]. The Cronbach’s alpha for this scale was 0.88. This instrument has been used in a United Kingdom healthcare worker sample during the COVID-19 pandemic [45].

### 2.4. Statistical Analyses

Participants with partial responses were included using pairwise deletion, with the omission of non-responses per variable rather than the implementation of missing value replacements. Data were analyzed with Mann–Whitney U and *t*-tests for continuous data, and Fisher’s exact and Chi-Square tests for categorical data. The Shapiro–Wilk test checked for normal distributions. Wilcoxon signed-rank and McNemar’s tests compared pre-pandemic and current data. Means, standard deviations, and ranges were calculated. Calculations and analyses were conducted using SAS 9.4 (SAS Institute Inc., Cary, NC, USA).

## 3. Results

Among the 114 participants that originally submitted the survey, 5 participant entries were removed due to lack of consent or nonresponse on all items, resulting in a total sample of 109 with at least partial responses. Of this total sample, 79.82% completed the entire survey, with similar completion rates between the prison (79.03% of 62) and community (80.85% of 47) groups. The analysis incorporated the remaining participants’ partial responses.

Table 1 shows the demographic characteristics of the study participants, with no significant differences between the two groups for gender, race, marital status, and age. The majority of participants in both groups were female, married or partnered, with a mean age within the 40-year age range. The largest racial subgroup was non-Hispanic White for both groups.

Table 2 indicates the organizational characteristics before and during the pandemic. Both prison and community nurses had mean work years of about 15 years, with a minimum of 2 years, without significant differences. The pre-pandemic and current weekly mean hours of work were significantly higher for prison nurses compared to community nurses. For prison nurses, work hours significantly increased during the pandemic. Although there were no significant differences in night shift assignment between the groups before the pandemic and currently, the amount of prison nurses working any night shifts significantly increased after the start of the pandemic. There were no significant differences in psychosocial work stress experiences in terms of effort-reward imbalance between the groups, but stress levels in both groups were relatively high (E-R ratio > 1.0).

Table 3 reports working conditions during COVID-19. Prison nurses reported significantly more direct COVID-19 patient contact, and had more requests to work, or had worked, in other departments. However, significantly more prison nurses perceived adequate PPE supply and had COVID-19 testing compared to community nurses. Significantly more community nurses expressed fear of contracting COVID-19 at work, and had a higher level of general fear towards the COVID-19 outbreak.

Table 4 focuses on the psychological wellbeing of the study participants. The prison nurses’ mean daily sleep hours were significantly lower than those of community nurses before the pandemic and currently. Sleep hours significantly decreased for both groups compared to before the pandemic. Mean scores for total insomnia and the sleep-related items indicating “trouble falling asleep” and “waking up at night” were elevated in the prison group compared to the community group, but these differences were not statistically significant. Both groups did not significantly differ in their mean PTSD scores, but both mean scores were above the cutoff score of 14. The percentage of nurses with a PTSD score equal to or above 14 was 49.02% in the prison group and 69.05% in the community group. Although depression and anxiety mean scores were more elevated in the community group, they did not significantly differ from those of the prison group. For both groups, the depression and anxiety mean scores were below the cutoff, and the prevalence of depression and anxiety cases was low.

## 4. Discussion

Significant findings from this study provide insight into prison nurses’ intensified challenges, including longer work hours, less sleep hours, more COVID-19 patient care demand, higher perceived PPE supply, and lower pandemic-related fear levels compared to community nurses. Although not statistically different, the occupational stress and mental distress results of prison nurses and community nurses are concerning, and reflect the pandemic context.

The weekly work hours of the prison nurse study participants contrasted with those of the U.S. nurse population. Among the estimated U.S. population, 58.7% of nurses worked 32 to 40 h weekly between February and June 2020 [1]. While the community group’s mean pre-pandemic and current weekly hours were within this range, those of the prison group exceeded the national population estimate. Additionally, this finding of long working hours among the prison nurse study participants may be related to the previously mentioned issue of mandated overtime among some California state institutions [34]. The World Health Organization and International Labor Organization have confirmed that long working hours of 55 or more weekly hours are an occupational risk factor for cardiovascular disease [46].

Our study’s findings on sleep hours and related issues align with a pre-pandemic Washington state prison study, in which the majority of correctional staff and healthcare workers had less than five hours of sleep, 53% had zero to two hours of sleep between work shifts, and 40.7% to 47.2% of participants reported over thrice-weekly trouble falling asleep and nightly waking [47]. In contrast, among East U.S. correctional healthcare workers during the pandemic, 81.57% did not have sleep disturbances, indicated by normal score ranges, but their mean sleep disturbance scores were significantly higher compared to correctional officer mean scores [18].

The prison nurse study participants experienced higher pandemic-related work demand, schedule, and environment adjustments through increased COVID-19 patient care, increased night shift assignments, and increased department redeployment requests. The elevated California correctional facility COVID-19 case rate [21] may have contributed to these occupational changes. Furthermore, a 2022 report from the National Commission on Correctional Health Care featured the ongoing issue of understaffing among certified correctional health professionals throughout the two years of the pandemic [48]. COVID-19 patient contact, shift assignment, and department redeployment work changes were similarly identified as challenges in the pandemic experience of U.S. hospital nurses [32]. COVID-19 patient care has been associated with increased risk of infection, anxiety, and emotional distress among nurses working in hospitals and other clinical settings [49,50,51].

The perception of adequate PPE supply may have been related to the lower degrees of COVID-19 general and work infection fears, despite significantly increased COVID-19 patient care among prison nurses. In contrast, among community nurses, there was a higher proportion that perceived inadequate PPE supply, elevated COVID-19-related fears, and a lower proportion of COVID-19 patient care. The perception of adequate PPE availability has been previously associated with a reduced probability of COVID-19 infection among U.S. healthcare workers across multiple clinical settings [49].

The history of the CDCR’s mandatory overtime practice [34] and our prison nurse study participants’ long working hours and COVID-19-related work changes exemplify the influence of occupational policy on nurses’ health and wellbeing, as well as highlight implications for organizational interventions. A scoping review of nurses’ coping during COVID-19 emphasized the importance of work hour and schedule flexibility and occupational safety via PPE and training [52], encompassing organizational responsibilities.

Considering our findings of elevated work hours and COVID-19 patient care, lowered sleep hours, and sleep quality issues, our sample of prison nurses may be at risk of high fatigue. There have been associations of increased work hours and COVID-19 patient care with increased fatigue among hospital nurses working through the pandemic [29].

Although other nurse- and work-related stress instruments have been used within this population before the pandemic [14,15], the ERI instrument has not been previously applied to correctional nurses to measure occupational stress. The similar mean E–R ratio scores among the prison and community nurses corroborate previous ERI results from other nurse and healthcare worker populations working through the pandemic. Italian hospital healthcare workers had a mean E–R ratio over 1 [22], and three-fourths of nurses working in Greek hospitals scored E–R ratios over 1, demonstrating occupational stress in the context of the COVID-19 pandemic [23]. In contrast, pre-pandemic studies among correctional officers in China and general nurses in Europe indicated a mean E–R ratio score under 1 [38,53]. With respect to the findings of our study, it is suggested that the current pandemic circumstances may contribute to increased occupational stress in all types of nurses.

Addressing the high levels of occupational stress suggested by both nurse groups may imply feasible approaches with symptom reduction interventions. The published reviews of individual-based nurse intervention studies have identified evidence of mindfulness interventions as being helpful in the reduction of work stress and burnout [54], as well as the prominent use of technological mediums to promote intervention accessibility [55]. Therefore, potential interventions to reduce stress among nurses working in prisons and within the community may involve mindfulness techniques administered through a digital format.

Regarding psychological measures, the high prevalence of PTSD symptoms in our study contrasts with the findings of previous studies on correctional healthcare workers. Pre-pandemic Canadian studies that categorized nurses within a minority subgroup of correctional wellness workers found a lower prevalence of positive PTSD screens, at 16.7% and 17.2%, which suggests the pandemic’s effect on correctional healthcare worker post-traumatic stress symptoms [16,17]. Considering the COVID-19 pandemic, our study’s psychological findings differ from those of a recent East U.S. correctional worker study. Almost half of our prison nurse group had positive PTSD screens, while in the latter study, 43.27% of correctional healthcare workers had normal PTSD scores and 20.29% had moderate and severe scores [18]. Both our prison group and the East U.S. healthcare worker subgroup had a low prevalence of depression and anxiety screens [18]. The prevalence contrasts among these studies may be due to differences in instrumentation, timing, and setting. Although the scope of our study may not provide causal inferences, our prison group’s higher prevalence of PTSD may be related to our single-site focus and a previous finding among hospital nurses of significantly elevated PTSD scores associated with COVID-19 patient care [29].

This study had the following limitations, including its relatively small sample size. Our findings from one California prison may not be generalizable to all California prisons, nor the whole correctional nurse population. Although most of the nurses in the community group were from California and worked in hospital settings, the heterogeneity of this group may present a nonequivalent comparison to the single-site prison nurse group due to potential differences in the work settings. The extended and nonoverlapping survey windows of the two groups also present the possibility of different historical effects affecting each group. The self-report nature of the survey and the retrospective items referring to before the pandemic may have introduced recall bias. Regarding the possible generalizability between our study participants and the correctional and general U.S. nurse populations, the demographics of our study participants reflect those of the 2020 U.S. nurse population estimates, with a female and White majority [1]. However, the sample and population estimate distributions underrepresent male nurses and nurses of other ethnicities, warranting future research to investigate the occupational experiences and needs of these minority groups.

The study’s strengths include its timing within the first year of the COVID-19 pandemic, which provides data of the pandemic’s initial impact on the prison and community healthcare settings. Despite limitations from the study design decision to retroactively add the prison nurse group, this addition contributes to the representation of the correctional nurse population. Likewise, the allowance of one- to two-month survey windows and the use of short-form versions of validated instruments were intended to accommodate participants’ time and stress while working during this phase of the COVID-19 pandemic. The comparative study design between a specialized nurse group and a general nurse group, rather than a different occupation, highlights the needs and circumstances of prison nurses. The participants in both study groups were experienced nurses, with a minimum of 2 years and a mean of about 15 years of work, which avoids potential confounding from newer nurses transitioning into the practice. This study addresses the geographical and nursing specialty knowledge gaps by providing data on Californian prison nurses. Although the prison nurse group sample size was small, our study may be considered a pilot study, contributing knowledge related to the COVID-19 pandemic in the context of the need for data on the correctional nursing workforce [2].

## 5. Conclusions

Our group of prison nurse participants significantly differed from the community nurses, with longer work hours, fewer sleep hours, higher COVID-19 patient care, higher perceptions of adequate PPE supply, and lower pandemic-related fear levels. Occupational stress and mental distress impacted both groups of nurses. Our findings suggest the need for and importance of future research and practice to improve correctional nurse occupational wellbeing, and consequently, the care and wellbeing of incarcerated individuals. Although our study was limited to a relatively small sample at a single prison, our findings may be considered preliminary for future studies involving multiple correctional facilities.

Addressing excessive working hours and sleep deficiencies among correctional nurses may involve the revision of institution policies regarding mandatory overtime, staffing, and more equitable distributions of workload. Additional organizational support in both correctional and healthcare settings may include sufficient PPE supply and accessible interventions and resources to manage mental distress among nurses in the workplace.

## Figures and Tables

**Table 1 ijerph-19-10955-t001:** Sociodemographic data.

Variables	Prison (*n* = 62)		Community (*n* = 47)		*p*-Value
	*n* ^a^	%	*n* ^a^	%	
**Gender**					
Female	45	93.75	36	85.71	0.36 ^b^
Male	3	6.25	5	11.90	
Transgender	0	0	1	2.38	
**Race**					
Non-Hispanic White	18	35.29	16	38.10	0.41 ^b^
Non-Hispanic Black	3	5.88	4	9.52	
Non-Hispanic Asian	10	19.61	13	30.95	
Hispanic or Latino	16	31.37	7	16.67	
Other	4	7.84	2	4.76	
**Marital Status**					
Single	15	30.00	13	31.71	0.83 ^c^
Married or Partnered	28	56.00	24	58.54	
Separated or Divorced or Widowed	7	14.00	4	9.76	
**Age in Years (mean ± SD)**	44.29 ± 9.44		41.29 ± 12.78		0.12 ^d^

^a^ Sample sizes vary per variable due to missing data. ^b^ Fisher’s exact. ^c^ Chi-Square. ^d^ Mann–Whitney U.

**Table 2 ijerph-19-10955-t002:** Psychosocial and organizational work characteristics.

Variables	Prison		Community		*p*-Value
	Mean ± SD				
**Effort–Reward Imbalance**					
E–R Ratio	1.32 ± 0.44		1.28 ± 0.58		0.46 ^a^
Effort Score	9.62 ± 1.69		9.40 ± 2.06		0.65 ^a^
Reward Score	17.95 ± 3.82		18.73 ± 3.76		0.27 ^a^
**Work Years (mean ± SD, range)**	15.93 ± 10.52, 2–42		15.32 ± 12.49, 2–45		0.34 ^a^
**Pre-Pandemic Work Hours (weekly mean)**	41.51 ± 8.21		33.51 ± 13.95		<0.0001 ^a^
**Pandemic Work Hours (weekly mean)**	54.73 ± 14.52		33.51 ± 15.27		<0.0001 ^a^
**Pre-Pandemic vs. Pandemic Work Hours** **(*p*-values)**	<0.0001 ^b^		0.48 ^b^		
**Pre-Pandemic Night Shift**	***n* ^c^**	**%**	***n* ^c^**	**%**	0.61 ^d^
No	44	78.57	38	82.61	
Yes	12	21.43	8	17.39	
**Pandemic Night Shift**					
No	33	57.89	35	76.09	0.0526 ^d^
Yes	24	42.11	11	23.91	
**Pre-Pandemic vs. Pandemic Night Shift** **(*p*-values)**	0.0017 ^e^		<0.0001 ^e^		

^a^ Mann–Whitney U. ^b^ Wilcoxon signed rank. ^c^ Sample sizes vary per variable due to missing data. ^d^ Chi-Square. ^e^ McNemar’s Test.

**Table 3 ijerph-19-10955-t003:** COVID-19 characteristics.

Variables	Prison		Community		*p*-Value
	*n* ^a^	%	*n* ^a^	%	
**COVID-19 Patient Contact**					
Direct patient contact	53	91.38	24	52.17	<0.0001 ^b^
No direct patient contact but work with other HCW(s) who have direct patient(s)	5	8.62	12	26.09	
No direct patient contact but shared common spaces with other worker(s) and/or patient(s)	0	0	4	8.70	
No contact	0	0	6	13.04	
**PPE Supply**					
Adequate	46	77.97	25	54.35	0.0103 ^c^
Inadequate	13	22.03	21	45.65	
**Volunteered or Asked to Work in Other Department**					
Yes	54	87.10	19	40.43	<0.0001 ^c^
No	8	12.90	28	59.57	
**Fear of Work Infection**					
Strongly Agree	18	31.03	21	44.68	0.0115 ^b^
Agree	19	32.76	22	46.81	
Disagree	14	24.14	3	6.38	
Strongly Disagree	7	12.07	1	2.13	
**Fear of Outbreak (0–100, mean ± SD)**	51.80 ± 28.65		67.85 ± 22.96		0.0025 ^d^
**COVID-19 Symptoms**	***n* ^a^**	**%**	***n* ^a^**	**%**	0.92 ^c^
Yes	22	40.00	16	39.02	
No	33	60.00	25	60.98	
**COVID-19 Testing**					
Yes	55	100.00	20	46.51	<0.0001 ^c^
No	0	0	23	53.49	
**COVID-19 Diagnosis**					
Yes	7	12.73	3	6.98	0.51 ^b^
No	48	87.27	40	93.02	

HCWs = healthcare workers. ^a^ Sample sizes vary per variable due to missing data. ^b^ Fisher’s exact. ^c^ Chi-Square. ^d^ Mann–Whitney U.

**Table 4 ijerph-19-10955-t004:** Sleep and psychological characteristics.

Variables	Prison		Community		*p*-Value
	Mean ± SD				
**Pre-Pandemic Sleep (mean daily hours)**	6.60 ± 1.10		7.20 ± 1.13		0.0156 ^a^
**Pandemic Sleep (mean daily hours)**	5.36 ± 1.30		6.65 ± 1.51		<0.0001 ^a^
**Pre-Pandemic vs. Pandemic Sleep** **(*p*-values)**	<0.0001 ^b^		0.0099 ^b^		
**Insomnia Score**	6.20 ± 2.98		5.68 ± 3.45		0.44 ^a^
Trouble falling asleep	3.29 ± 1.08		2.98 ± 1.27		0.16 ^a^
Waking up at night	3.09 ± 1.14		2.86 ± 1.30		0.31 ^a^
Waking up too early	2.82 ± 1.25		2.84 ± 1.38		0.97 ^a^
**PTSD**					
Mean Score	14.57 ± 5.95		15.88 ± 5.42		0.17 ^a^
Score > 14 (*n* ^c^, %)	25	49.02	29	69.05	0.0514 ^d^
**Depression**					
Mean Score	1.24 ± 1.48		1.76 ± 1.51		0.06 ^a^
Score > 3 (*n* ^c^, %)	8	16.00	9	21.95	0.47 ^d^
**Anxiety**					
Mean Score	1.50 ± 1.61		2.02 ± 2.02		0.28 ^a^
Score > 3 (*n* ^c^, %)	9	18.75	11	26.83	0.36 ^d^

^a^ Mann–Whitney U. ^b^ Wilcoxon signed rank. ^c^ Sample sizes vary per variable due to missing data. ^d^ Chi-Square.

## Data Availability

The data presented in this study are available on request from the corresponding author. The data are not publicly available due to privacy reasons.

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
