# Peer review of "Working Conditions and Wellbeing among Prison Nurses during the COVID-19 Pandemic in Comparison to Community Nurses"

_ijerph, 2022, doi:10.3390/ijerph191710955_

Round 1

Reviewer 1 Report

 I would suggest to revaluate the introduction part. Even if it is very convenient to read due to its scarcity, some theoretical background seems missing. For instance, I miss the deeper theoretical derivation of the link between prison nurses, Covid-19, and PTSD, although PTSD symptoms seem to represent a key research question. In general, the declared aim of the presented study seems not clearly specified within the introduction section. Although the aim is formulated, it seems not profoundly derived. I think the introduction section could be enriched by more results regarding the declared study questions.

·        “Design”: I think that the phrasing “over the course of six months” is misleading, as it sounds like a longitudinal study. I would suggest to change the wording for more transparency. You might further explain the relatively wide time frame for a cross-sectional study that aims to examine a highly topical situation.

·        “setting and sample”: I would appreciate more details about the recruitment strategy (How did you choose the organizations? How was the drop out?, etc.)

·        “Discussion”: Do be honest, I don´t understand, why you mention that you did NOT conduct correlational analyses. I guess it could have be done if essential?

·        “Limitations”: I think that your study shows more limitations that need to be mentioned/explained/discussed: you use (among others) retrospective data (e.g., pre-pandemic sleep), you use ultra-brief instruments or single items from validated instruments, male participants are highly underrepresented and it would be good to have a guess if this represents the gender distribution within this specific population in general   

Reviewer 2 Report

This is a very interesting article with very important information to the effects of COVID-19 pandemic.

Is there any evidence why Californian prison nurses were more affected than nurses in other states? Is there lack of personnel, is there any issue of inefficiency in the prison administration?

Although the survey was comparative (prison nurses vs. community nurses) this was conducted only in one prison across the state of California. Authors should justify the reasons for that methodological approach. Otherwise, there is an issue of sample validity and reliability.

Why the ''prison nurse study participants experienced pandemic-related work demand, schedule, and environment adjustments through increased COVID-19 patient care, increased night shift assignments and increased department redeployment requests''? How other state prisons dealt with the issue?

Authors refer to "pre-pandemic Canadian studies in their discussion section but there is no such reference in their introduction. Presenting new information at the end of the paper might create confusion to the reader."

I’m not satisfied with the conclusions section.  It’s extremely limited and does not provide any rap up of the study and does not further elaborate on authors' suggestions.

Reviewer 3 Report

A small research group.

The discussion lacked references to other such studies - the discussion should be supplemented.

Conclusions is too general. Maybe because there are no statistically significant differences between the groups?

The main question regarding the working conditions of nurses in general, especially those working in prisons, is very important. However, the small research sample does not allow for broader conclusions. The study could be a prelude (pilot) to a larger-scale, multi-center study. The topic is important and original. It raises the need for further study of the topic. Yes, manuscript is clear and easy to read. As I wrote the review, the conclusions are too general, there are too few of them. They relate little to the main research problem and too general.
